# Optimal Exploitation of On-Street Parked Vehicles as Roadside Gateways for Social IoV—A Case of Kigali City

**Twahirwa Evariste [1],*, Willie Kasakula [1], James Rwigema [1] and Raja Datta [2],***

[1] African Center of Excellence in Internet of Things, University of Rwanda, Kigali 3900, Rwanda; Kasakula_219013718@stud.ur.ac.rw (W.K.); j.rwigema@ur.ac.rw (J.R.)

[2] Department of Electronics & Electrical Communication Engineering, Indian Institute of Technology, Kharagpur WB-721302, India

* Correspondence: e.twahirwa@ur.ac.rw (T.E.); rajadatta@ece.iitkgp.ernet.in (R.D.)

**Abstract:** Vehicular Ad Hoc Network (VANET) is a subclass of Mobile Ad Hoc Network that mainly consists of moving and/or stationary vehicles, connected through wireless protocols such as IEEE 802.11p and wireless access in vehicular environments (WAVE). With the evolution of the Internet of Things (IoT), ordinary VANET has turned to the Internet of Vehicles (IoV), with additional social aspects, a novel extension themed SIoV has become common in urban areas. However vehicular wireless communication paradigms exhibit short radio communication. This problem has always been approached by supplementing moving vehicles with stationary Road Side Infrastructures, commonly known as roadside units (RSUs). The penetration of such RSUs on the global market is very low; furthermore, their procurement, deployment, and maintenance costs are prohibitively very high. All mentioned challenges have discouraged the widespread deployment of roadside infrastructure especially within large urban scenarios. With this research, we leverage on-street parked vehicles to allow them to exist as temporal gateways in the case study area. A novel modeling technique is introduced to enable a specific Percentage of parked vehicles to take up the role of roadside gateways for a certain percentage of their parking time. A mobile application is implemented that manages parking duration of the vehicle, based on the arrival, and departure time frames. Two more existing strategies were discussed (road-intersection RSUs deployment approach and Inter-vehicle scheme) to validate our proposed method through comparative studies. To evaluate the network performance evaluation, we compare two performance metrics, that is, Packets success delivery rate, and overall packets throughput under numerous vehicle densities. Using parked vehicles as temporal roadside gateways has demonstrated better results in comparison to intersection based RSUs deployment approach, and free vehicle to vehicle communication approach.

**Keywords:** SIoV; IoV; IoT; VANET; OSM; SUMO

## 1. Introduction

In recent years there has been a steady growth in the Internet of Things (IoT), a novel paradigm that is being materialized by integrating physical objects based on several technologies such as Radio Frequency Identification, Machine to machine and sensory technologies [1,2]. The services it renders to numerous sectors including Smart cities [3], Smart Agriculture [4], Smart Manufacturing [5], among others, have significantly improved the lives of human beings in the above fields. For instance, IoT has improved the industrial revolution to industry 4.0 which makes it possible to carry out production line based monitoring that transforms traditional industrial processing lines [6].

The role of internet of Things (IoT) is to equip physical objects with computing, processing and communicating capabilities so that they can interact with each other over an established network, IoT enables multidimensional technological approach that brings together Ubiquitous technology, pervasive technology, Internet Protocol, sensing technologies, communication technologies, and embedded devices, both physical, and virtual worlds can interact [7]. Internet of Vehicles (IoVs) is a form of IoT where vehicles are the communicating objects, IoV has been realized from vehicular Ad Hoc Network that has been in existence for many decades [8]. IoV has demonstrated substantial impacts on smart cities using the remarkable advancements observed in information communication technologies. Social Internet of Things (SIoT) is an extension of IoT where connected objects form a network through which they exchange information, especially in the social aspects [9]. SIoT communication technologies have been developed, and the security of nodes participating in the social internet of things has been a great concern as well. The authors of Reference [10] introduce a protection scheme that secures the location of the source to address the privacy source node location issue, and privacy disclosure attacks problems could be resolved.

SIoT has introduced enhanced interactions between human and physical objects by improving network extensible, reliability, and resource discovery. For example, the novel Social web of things (SWoT) was introduced that facilitates human beings to interact online with real-world physical entities [11]. With SIoT, interaction among humans is possibly applicable to real-world objects that exhibit ubiquitous computing [12]. Objects participating in SIoT network autonomously communicate with each other based on the owner's rules. Social Internet of Vehicles (SIoVs) is the novel theme of SIoT where vehicles are the major entities that form a network, this network allows the chauffeured vehicles and their drivers have social interactions.

At present, rural-to-urban migration is steadily increasing and this significantly leads to urban growth in terms of development and population. The advancement in Intelligent Transport System (ITS) has proven its integral contribution to smart-city development, transportation essential role in the modern lives has gone beyond enabling urban vehicular mobility practices but also assist people in these cities [13], especially in the aspects of trip planning. Vehicular Ad Hoc Network (VANET) is the most influential component of ITS, based on the ordinary VANET advanced vehicular technologies like vehicular cloud computing, vehicular fog computing could be implemented [14], and the problems associated with computational complexity, big data processing, bandwidth consumption could be resolved [15]. Through numerous protocols such as IEEE 802.11p and Wireless Access in Vehicular Environments (WAVE), VANET enables vehicles to communicate among themselves and valuable messages are exchanged. There exists challenges also associated with IoT data integration, indexing, and management from multiple sources [16], even-though research efforts have been put forward to address such issues, handling massive data from multiple IoT sensing devices like internet of vehicles still needs efforts in terms of industrial development and research. Ensuring mobility context-aware representation in IoV, and privacy of data from multiple sensing devices that are common features of IoV is very crucial.

Without losing generality, we categorize VANET communication modes into two main communication modes, that is, Vehicle-to-vehicle (V2V) and Vehicles-to-Roadside Units (V2R) communication. However, with the introduction of new paradigms such as the Internet of roads, VANET allows vehicle-to-everything communication (V2X) which involves Vehicle-to-Infrastructure (V2I), Vehicle-to-Device (V2D), Vehicles-to-Grid (V2D), Vehicle-to-Pedestrian (V2P), vehicle-to-Broadband cloud (V2B), generally speaking, V2X focuses on safety, traffic flow efficiency, social, or infotainment applications [17]. VANET requires a certain number of vehicles to ensure good connection and functionality, sometimes this becomes impossible; either due to the insufficient number of vehicles in the area of interest or unwilling to participate in the vehicular network by some individuals. But most importantly, it may be as a result of very few vehicles with on-board units. The most adopted approach to resolve the mentioned issue is to support V2V with V2I, specifically by deploying alongside road static infrastructures (RSUs). However, the confluence

expenditure on both procurement and servicing RSUs has become prohibitively very high in the aspects of large scale deployments, despite the greater supplement it renders to VANETs.

In this section, we introduce a typical SIoV system architecture that comprises of four (4) main layers, Figure 1. Layer 1 is the perception layer which involves all real-world physical entities such as vehicles, drivers, passengers, pedestrians, Roadside infrastructures, and so forth. This layer facilitates data sensing and acquisition from both intra-vehicle and environmental sensors, most importantly data may also come from the devices held by passengers, chauffeurs, and other individuals that are part of the integral SIoV ecosystem. The network layer plays a vital role to interface perception layer and cloud-based framework. The sensed data is transmitted to the edge computing layer which brings the cloud services very closer to the users. Whereas the edge computing layer pulls computation, data storage closer to the end-users, in other words, it brings the cloud computing at the edge of the network so that the problems associated with latency, large bandwidth consumption will be resolved. For more enhanced delay-sensitive, processing, and communication requirements, Fog computing is introduced at this level to ensure enough storage of substantial data from a vast number of fog computing nodes. Last but not the least is the cloud computing layer that employs cloud technologies to avail very large storage capacities and complex computations are handled at this layer. All shared social aspects for the entire SIoV architecture are handled at this layer. Sensing, Platforms, Storage, Infrastructure are used as services based on the system requirements.

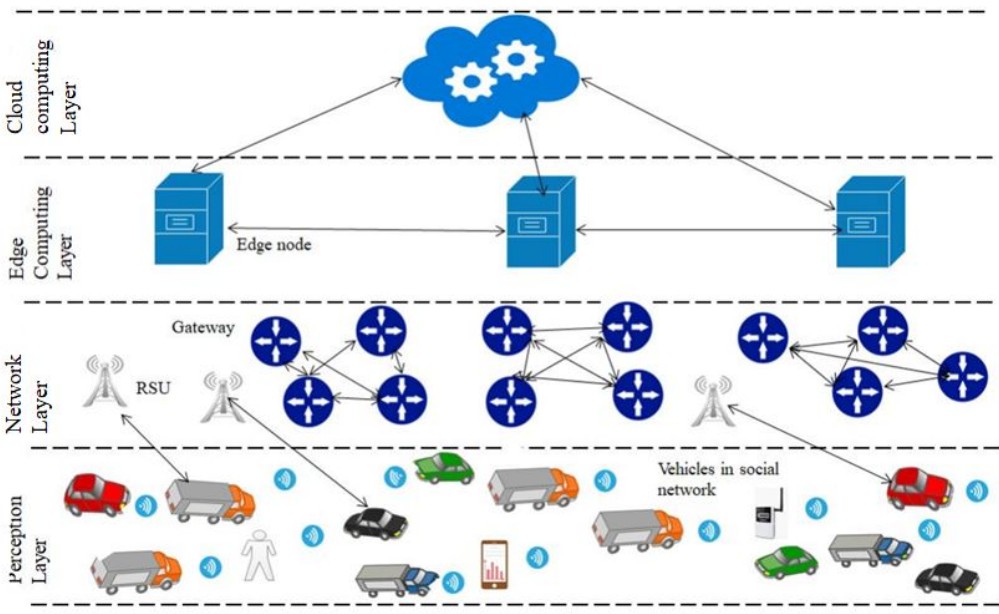

**Figure 1.** Layered Architecture of Social Internet of Vehicle (SIoV).

With this research, we intend to optimally exploit the parked cars as temporal roadside gateways to enhance the performance of the Internet of vehicles and allow possibilities in vehicles' social interaction. The on-street parked vehicles situation is naturally possible in almost all cities, and a good number of vehicles stays for sufficient time during the day. Figure 2, describes a combination of two traffic modes, that is, Parked, and mobile vehicles. Only a specific percentage of parked vehicles are involved in connecting moving vehicles. Interestingly, parked vehicles could relay messages to the intended recipients out of their ranges through other on-street parked vehicles.

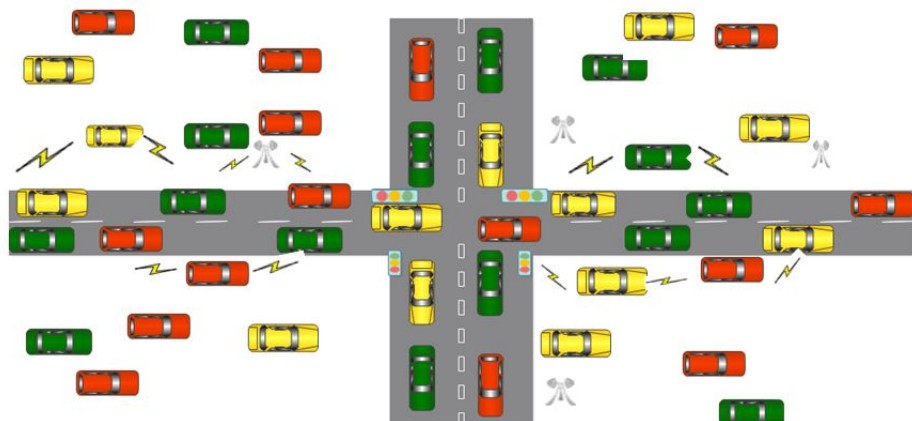

**Figure 2.** Reference Scenario.

Not many research works have taken into consideration the importance of parked cars and their valuable roles in IoV. Some vehicles are not good candidates at some parking sites due to their parking duration and based on the fact that some when are parked, their engines are off and so does their electronics. We, therefore, show leveraging a certain percentage of parked vehicles for a specific percentage of their parking duration, with rechargeable onboard electronic components. A better vehicular network could be set up consisting of both mobile and parked vehicles. To ensure realism, we have utilized real parking information data from various sites in Kigali city, including Commercial, Administrative, and residential parts.

Real maps of the target areas have been obtained from Open Street Maps (OSM), and mobility simulations were carried out using Simulation of urban mobility environment (SUMO) to generate road traffics. With this information, we come up with a model that optimizes the utilization of parked vehicles as temporal roadside gateways which would conserve a bigger number of units that would contribute based on the traffic flows augmentation. Three (3) Approaches have been considered in this research work, and these include:

- RSUs enabled Strategy at road intersections: A common vehicular infrastructure deployment strategy where infrastructure are placed at some hotpots road intersections for numerous vehicular applications such as, traffic control and monitoring, driving mistakes monitoring, and so forth.
- Inter-vehicle-communication: This is typical vehicle-to-vehicle (V-2-V) communication, where vehicles exchange messages without the intervention of any deployed infrastructure.
- Parked vehicles approach. In the parked vehicles approach, we utilize a specific percentage of the on-street parked vehicles as temporal gateways from the identified on-street parking sites.

Results show that our method performs better than the rest based on renowned Network performance metrics (i.e., Overall Network Throughput, % Packet Delivery Ratio) for all areas under consideration, that is, Administrative, commercial, and residential. We summarize the main contributions of this work below.

1. An android based Parking management application (PMApp) was designed and implemented that enabled vehicle parking data collection.
2. Based on the real parking traffic information from the target area, we introduce a model that exploits parked vehicles as roadside gateways for SIoV.
3. A simplistic LoRa IoT-based gateway prototype is implemented and tested using Lora compliance sensing nodes. The experimental results show the prototype's capabilities of accommodating several computational tasks.

4. Network simulation studies were conducted to evaluate the network performance effectiveness. And numerical results present on-street parked vehicles as rich temporal infrastructures for vehicular networking in urban scenarios.
5. A novel Parking matrix is introduced that holds all the information regarding parked vehicles. The length of time each vehicle spends in each parking area is recorded.

The remainder of this paper is organized as follows. Section 2 discusses the works that are related to ours. In section 3 we present the methodology, overall research design, area of interest demonstration, and data collection from the same are. Parked vehicle relocation models are described in Section 4 and further discuss the proposed strategy in detail. Section 5 enlights the open innovation engineering in optimal exploitation of On-street parked vehicles. The analysis of both experimental, and simulation results is presented in Section 6. Section 7 concludes the paper, and shapes our future research.

## 2. Related Work

There have been tremendous efforts on vehicular communications research works, since the previous years ranging from traditional vehicular Ad Hoc Network to its novel extension themed Internet of vehicles, and social internet of vehicles. Originally, the social network of vehicles was introduced to assist vehicles exchange safety, efficiency, and comfort-related messages. The Social Internet of vehicles has evolved from the current internet of vehicles that equip vehicles with cognizant, and onboard sensing capabilities [18]. Vehicles can recognize their environments, and share their context with peer vehicles, in a single, or multihop environment. On-board context-aware of vehicles has been a focus as a building block for vehicular Ad hoc Networks [19].

Generally speaking, vehicles in SIoV interact among themselves to share data on road conditions, identify heavy traffic routes, infotainment such as games, videos, sharing information about free parking lots, and so forth. The ethical, and moral implications architecture for SIoV is asserted in Reference [20]. Currently, research works regarding the social internet of vehicles tackle various dimensions of the vehicular technologies. Concepts, architecture, and applications of SIoV were proposed in Reference [21], the authors strengthen the concept of the vehicular cloud technology, base on it and propose cyber-physical architecture for SIoVs. A clear comparison is also presented that mark a line between the social network of humans, and social network of vehicles, the proposed architecture, supports social interaction among real-world physical entities to stimulate numerous forms of communications and information storage.

Extensible SIoV architecture based on web-based technologies is suggested in Reference [22]. SoIV user applications could be easily developed, additional enabling technologies, and social internet of vehicle protocols are highlighted as well. A cross-layer protocol that manages traffic in the social internet of vehicles is presented in Reference [23]. The authors introduce an analytical model for intelligent traffic congestion control based on the social internet of vehicles. A three-tier layered model is introduced, with physical entities that deal with sensory activities, cyber entity, for security services, and social graph for information storage.

Furthermore, a crowdsensing SIoV is studied in Reference [24], that enables real-time traffic control system in urban scenarios. Important components that are required for the proper implementation of network structures in urban social internet of vehicles are highlighted, most importantly, smart terminals, smart vehicles, RSUs, gateways, base stations, and so forth. Leveraging heterogeneity in SoIV networking is essential, and challenging. Traffic management server presented in this work, timely takes actions towards some road incidences like road accidents, road traffic congestions, and so forth. The advancement in technology, and social aspects, have seamlessly boosted the development of social internet of vehicles, and possibilities to build up intelligent techniques that enable the adoption of SIoV. The work in Reference [25] establishes vehicle path selection by using game evolution technique in the social internet of vehicles. Traffic flow congestion is minimized and controlled through establishing

optimal routes for vehicles. Vehicles under consideration are clustered and classified based on the social correlations among them via historical and current driving data.

The emergence of SIoV is based on the growing internet of things, and the internet of vehicles. The later encourages pervasive data exchange, and information sharing among vehicles themselves. It requires very little, or no human involvement. There is no doubt that the challenge of big data volumes associated with IoT, and IoV will affect the operation of SIoV in widespread environments, such as urban scenarios. Reference [26] learns a social big data-based information circulation technique that offloads vehicle-to-infrastructure data through device-to-device, vehicle-to-vehicle communication links.

Besides, vehicular networks deal with critical multimedia, real-time, and effective data that could warn overspeeding drivers, mitigating accidents from occurring, providing precisely the current position of the vehicle, and so forth. Data related studies for effective vehicular communications are very crucial to ensure all that mentioned above, a data-driven Clustering for Multimedia data communication is studied in Reference [27], the work optimizes data sharing in IoV through clustering vehicles, and data sharing is maximized among vehicles that interact with each other.

Modeling techniques that combine both moving, and stationary vehicles have been put forward to enhance the performance of the internet of vehicles, Reference [28] builds a vehicular system that controls traffics in urban areas where vehicles in the RSUs proximity are modeled as fog nodes. The proposed architecture is employed to solve the offloading optimization problem, computation complexity, very low latency could be guaranteed. A tnote social network infrastructure for both static Home Base unit, and dynamic vehicles on-board units was proposed by Reference [29]. An impressive architecture consists of radio communication units (OBUs, RSUs), tnote cloud, and user application is presented in this work.

Even-though SoIV builds on the existing vehicular Adhoc network, and internet of things, there exist several challenges that still hinder the initial deployment, and operational SIoV, these include context-awareness representation, high mobility, initial cost setup, and so forth. Furthermore, privacy preservation in the social internet of vehicles is realized as the potential challenge as well [30]. However, it is still problematic for pure inter-vehicle communication to withstand the coverage and connectivity challenges. This results from VANET short radio coverage and signal obstruction by urban area fixed features.

Roadside Units are therefore employed in the areas of interest to extend the ability of VANET. As stated in the previous section, the joint cost of both procurement and maintenance of RSUs discourages the infrastructure planners from materializing such a solution, the U.S Department of Transportation, reported USD 13,000–15,000 for procuring a single RSU, and USD 2400 for annual maintenance of a simplistic RSU [31]. This explains why RSUs deployment over a widespread urban area becomes an issue. Numerous techniques have been studied to optimally exploit installed roadside units for the internet of vehicles. Reference [32] studies a cost-effective method that allows sharing roadside infrastructure for the Internet of vehicles, to reduce the cost of expenditures that governments spend on RSUs. Advising the sharing of local government deployed RSUs for safety reasons to be used at the same time by other service providers, to cut the cost of several infrastructure deployments.

Conceptualizing on-street parked vehicles as temporal vehicular communication agents in urban scenarios has been proposed, equipping parked vehicles with the capability to communicate has drastically been evolving ranging from ordinary rsus to centralized radio access network, a highly flexible cost-effective approach is proposed in Reference [33] that supports wide urban areas. The work proposes an idea of utilizing parked cars as RSUs for smart cities. Parked vehicles have been enabled to back the moving vehicles through collecting, aggregating, and disseminating critical messages. In Reference [34], authors suggest leveraging parked cars as urban self-organizing road-side units. Their work assumes parked cars to self-organize and perform as a support network to the already installed urban infrastructures, vehicles are exploited opportunistically to support the roadside units,

or fully take up their duties and temporally exhibit roadside duties. The authors' score based decision algorithm determines whether a specific parked car acts as a RSU or remains in the sleep mode.

While Reference [35] defines parked cars as excellent RSUs, the work introduces a self-organizing network strategy that allows parked cars to generate coverage maps taking into consideration the received Signal strength as the major factor to make important decisions, a small fraction of urban parked vehicles extend the vehicular coverage. The work in Reference [36] asserts parked vehicle computing architecture that explore energy-efficient internet of vehicles. Scheduling computational resources is based on the vehicle parking periods, and energy utilization. Parked vehicles with idle resources are the higher candidates to be allocated more tasks for execution. Authors emphasize that parked vehicles hold enough, and underutilized computational resources that would be exploited for vehicular computation activities.

The authors of Reference [37] suggest a content downloading method with the help of Roadside cars for Vehicular Ad hoc network. In this research, an efficient content downloading approach is suggested with the assistance of stationary vehicles forming a number of clusters; the vehicles in clusters assist to download the content requested, and pass them to the vehicles in need of it, most especially mobile vehicles. Whereas in Reference [38], a vehicular content distribution method using parked vehicles is proposed through sharing at Roadside. A ParkCast approach with no investment is proposed to leverage alongside parked cars to disseminate content in Urban Vehicular Ad Hoc Network. The content is distributed among other vehicles that travel through the parked ones. Their approach yields a good performance in the distribution of contents with numerous traffic sizes.

The existing body of knowledge on the vehicular communications in urban scenarios has demonstrated capabilities that can be introduced by employing on-street parked vehicles as RSUs. Optimal models are still needed to wisely ensure the implementation procedures. In this paper, we propose an optimal exploitation of on-street parked vehicles to ensure a specific percentage of vehicles acting as roadside gateways for a specific percentage of their parking duration. Other VANET enabled techniques are studied for comparison purposes. We firstly, carry out an RSU-based approach placed at the prominent road intersections, then unaided inter-vehicle scheme. We show that the benefits of parked cars surpass other methods under various vehicle densities. And proves to withstand the ever-increasing requests from increased mobile vehicles which is the real case in urban areas. In the next section, we describe the methodology of this study, the research design, and the techniques used for the data collection.

## 3. Methodology, Research Design, and Data Collection

In this section, we discuss the research method that was used for this study. Weekly traffic parking data is collected from three categories of the target city area, and described as administrative, commercial, and residential, based on the dominating business in an area. A specific percentage of the parked vehicles take up the role of roadside access points to allow vehicles with common social interest exchange messages. After all, in the IoV vehicles need to communicate and relay messages to the intended recipients via roadside infrastructures, the Simulation studies were performed to evaluate the network performance subjected to numerous vehicle densities in the area of study.

### 3.1. Simulation Design

Here we discuss the network performance simulation design and traffic data collection in the target area. Three important steps are taken to evaluate the network performance regarding vehicle speeds. Firstly a target area is extracted from an open street map (OSM) with road details, secondly, the OSM map version is converted to a microscopic urban mobility model known as the simulation of urban mobility (SUMO) [39]. Mobility files are obtained that demonstrate the travel details of all vehicles in terms of their identification (Node number), geographical coordinates of their travel routes for mobile vehicles, and coordinates of their parking site for stationary vehicles, and parking times in seconds.

Last but not the least, the mobility files are fed to network simulator version 3 (NS3) [40] to observe overall inter-vehicle and vehicle to roadside infrastructure communication. Although the network performance evaluation is simulated, to ensure the realism of our output in this study, we have considered a real area of study and real traffic flows. The traffic data of parked vehicles utilized was collected on a full week from the target area. Subsequently, the output of one software is conveniently input to another software. This has motivated us to select the three software applications(OSM, SUMO, NS3) for our simulation study as presented in Figure 3. In addition to that, all utilized software applications are free and open source. Feeding OSM to SUMO results road-network files, and mobility files are created. Vehicles mobility files are fed to the Network simulator to learn how vehicles in motion communicate and how they communicate to the peer vehicles in respective on-street parking lots. We make use of python scripts to extract mobility files from the macroscopic urban mobility simulation platform. Real urban entities are represented such as roads, buildings, parks, gardens, and so forth. In the microscopic sumo modeling, we simulate the mobility of each single vehicle and mark the geographical positions of the parked vehicle. While the SUMO road networks involve both main, and residential roads. In this study, we consider only the main roads that experience heavy traffic flows daily.

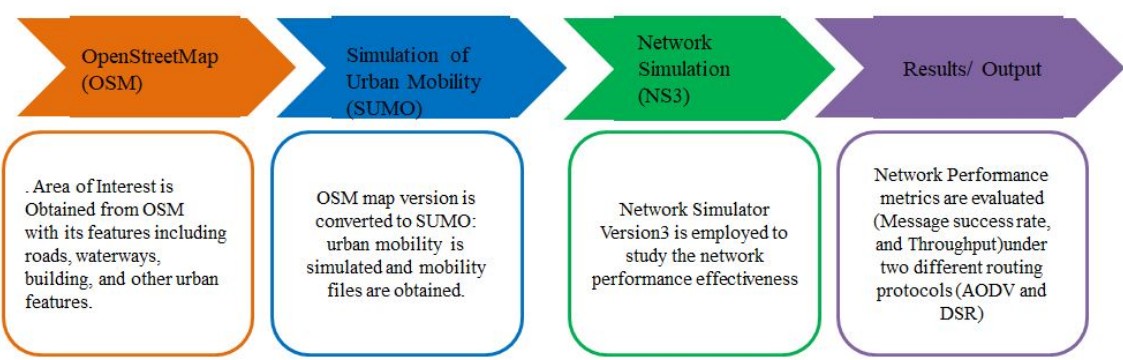

**Figure 3.** Simulation Design.

The possibility of a vehicle changing parking area is also considered in this study. For instance, a vehicle $V_1$ could visit three 3 parking areas in a single day. Mobility files with possible positions of the parked vehicles are seamlessly convenient to the network simulation tools like Network simulator version three (NS3), OMNET++. In this work $NS_{3.6}$ is employed to simulate the network performance effectiveness, it is a network simulator that is based on C++.

In the next section, we study a model that allow newly arriving vehicles to participate in a social message relaying for a specific percentage of its parking time. Interestingly, the common on-street parking sites are known, and the average number of vehicles could also be estimated based on the weekly data at hand. Equipping the available vehicles with access points devices would be a promising replacement of excessive required roadside infrastructures that consumes deployment budgets, as well as urban land that could be used for other purposes.

*3.2. Area of Interest Representation*

The region of interest is obtained from Open Street Map that contains a collection of roads and on street parking lots [41]. Based on the nature of dominance of businesses conducted in the area under consideration, we divide it into commercial, Administrative and Residential areas. According to the settlements in the city. To ensure realism, we have considered the actual traffic sizes based on the weekly data collection. This has given us the traffic density variation during week days and weekends.

Figure 4a–c indicate the open street map versions of the target areas, and potential parking areas in all administrative, commercial, and residential regions under study. OSM as a geo-data repository, avails important features that facilitate geographical map simulations. During traffic

parking data collection, 65, 78, and 30 potential parking areas were identified in the administrative, commercial, and residential areas, respectively. Each parking area is represented by its geographical position in terms of latitude, longitude geographical coordinates. Whilst Figure 5a–c demonstrate the Simulation of urban mobility (SUMO) map versions of the target area. The SUMO maps play vital roles in simulating the mobile vehicles that are connected by the stationary gateways supported in the parked vehicles, the positions of vehicles in the optimal parking lots are considered during network performance studies.

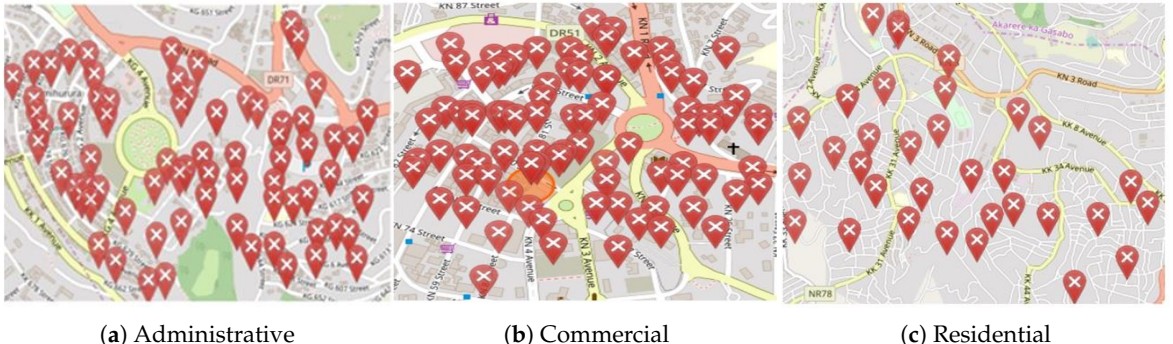

| (**a**) Administrative | (**b**) Commercial | (**c**) Residential |

**Figure 4.** Area of Interest representation: OSM version.

The convenience of utilizing SUMO mobility files for network performance evaluation studies motivates us to opt for the combination of these open-source software, and we define them in an acronym OSN (**O**penstreetMap, **S**umo, **N**S3). We employ python scripts to randomly estimate the positions of both mobile and static vehicles under consideration. Furthermore, the continuous availability of parked vehicles throughout the day has motivated us to exploit them as temporal stationary infrastructures. During simulation studies, some assumptions are made. Each parked vehicle at least lasts 100 s in the parking areas.

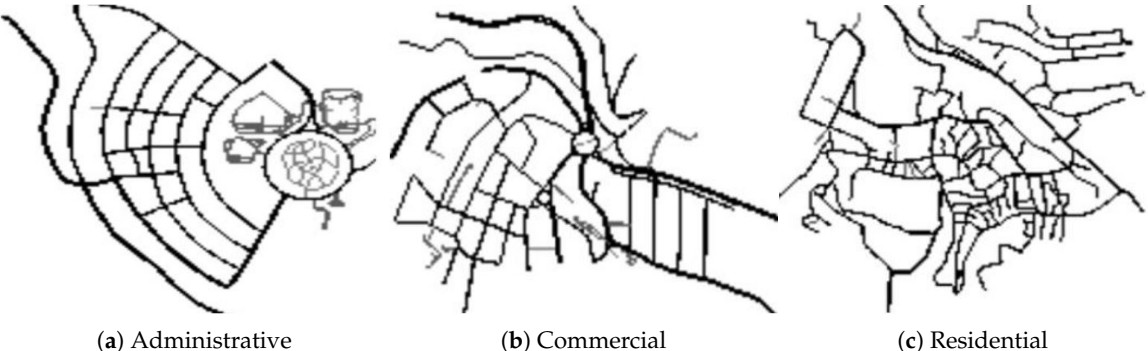

| (**a**) Administrative | (**b**) Commercial | (**c**) Residential |

**Figure 5.** Area of Interest representation: SUMO version.

In this simulation study, we assume vehicles move in a bidirectional lane, and all vehicles in both directions can communicate to the parked vehicles close to their itineraries. Since we model vehicles changing parking position. We assume that a vehicle could change position on the same parking, or change its position to a new street. The parking time in each position is recorded, and its participation in communication in each position is considered for each position. The traffic fluctuations in the target area is overwhelming, especially in the ways traffic volumes augment. Exploiting their presence could serve as a durable solution for vehicular communication in the target area. While the on-board electronics of the vehicles are powered by the vehicle's engines while in motion, it is essential to render them with renewable power sources to enable the access points while vehicles are parked.

At present numerous communication protocols exist that facilitate vehicular communications, and these include but not limited to Dedicated short-range communications (DSRC) that support the short-range radio communications common in vehicular networks [42]. Long Range Low power (LoRaWAN) area network has not been widely adopted in vehicular networks, but is expected to be the best candidate for future internet of vehicles [43], due to its two (2) attributes that vehicular communications mostly require, that is, able to propagate over long distances and low power consumption. Besides, it associates between Internet Protocol version 6 (IPV6) that supports excess load, a lot of data could be transmitted between a moving object like a vehicle, and internet node, like a stationary roadside infrastructure, high-level applications are supported [44]. Narrow Band IoT has been introduced to the internet of vehicles, and could further help in social-related data collection, and analysis. Narrow Band Internet of vehicles is also a Low Power Wide Area Network (LPWAN) which could fit well in vehicular communication circumstances [45]. Due to the ever-increasing voluminous vehicular data, the need of higher data rate and very low latency communications in vehicular networks, the next-generation cellular network (5G) is finding its way in this communication paradigm due to its nature in solving the problems mentioned [46]. Accessing data from various sources and providing end-users with proper integrated results is challenging, and defining access control policies has always been a concern of researchers [47].

In the communication simulations, we employed IEEE 802.11p for vehicular communications. While random mobility files are considered for network simulations, to ensure a more realistic situation, we employed raw traffic vehicle parking data to conduct the study.

### 3.3. Parking Management Application (Pmapp)

In this section, we introduce an intuitive mobile application that is employed to estimate the parking duration of each vehicle that visits the parking lots under consideration. It is important to know the amount of time each car stays parked in a particular parking lot. This can help in the determination of what car to deploy infrastructure to aid with the connectivity in Vehicular Adhoc Networks. The main aim of this app is to collect data on what car parked in which parking lot and the time it parked and left the parking lot. The app was developed using Java and an SQLite Database connected using the Room Persistence Library. The Figure 6 describes the architecture of the application. When users launch the app, they are taken to the home fragment where they have two options provided by two buttons as follows:

(a) Arrivals:

When this option is chosen, the user is taken to a fragment where they are provided with fields to enter the following details: (i) Number Plate (Text Field), (ii) Area Type (Dropdown List), (iii) Parking Lot (Dropdown List), (iv) Arrival Time (Time Picker Dialog). When the information is provided, the user can press the save button which proceeds with persisting the information in the database. The user is then redirected back to the home fragment.

(b) Departures:

When the option is selected, the user is taken to another fragment which displays a list of all parking events with no corresponding 'leaving time'. The user can then press on one of the events which takes them to another page where they are provided with a 'leaving time' field to enter the leaving time of the selected parked car: i. Leaving time (Time Picker Dialog) When the time is entered, the user can then press the save button which proceeds to persist the information in the database. The user is then redirected back to the home fragment.

The main interface of the app is indicated in Figure 7a, the PTApp manages two important operations that influence the length of time that each vehicle spends in a specific parking lot, that is, arriving at and departing the lot (see Figure 7b,c), respectively. Each parking vehicle's number plate is registered and the area type (Administrative, Commercial, and Residential). Both Parking area arrival and departure times are seamlessly added by the android calendar. The importance of the application is to easily collect the difference of the two-time slots, and this has given us information relating the length of time each vehicle spends in a specific parking area. Parking tolls collection will be supported by the application, and reports could simultaneously update both the car driver and parking owner, this will also increase trust between two. However, the scope of the application in this particular study is to compute the length of time that each vehicle spends in a specific parking area.

Collected information is automatically saved in a structural database at the back end, and used for modeling purposes to demonstrate the importance of the parked vehicles. The instantaneous data record consists of the 'NUMBER PLATE', 'PARK NAME', 'AREA TYPE', 'PARKING DURATION'. The last column information is extracted from both the arrival and departure times of each vehicle. In the next subsection, we highlight the parking traffic distribution in the three areas collected on a full-week basis. In the era of smart cities that require many computational resources, parked vehicles have been identified as the urban distributed computational resources that could serve peer mobile vehicles in their proximity. In the next subsection, we describe the on-street parking traffics in three areas of the city under consideration. The areas are named administrative, commercial, and residential. The names are derived from the nature of business and settlements in the these areas. Parked vehicle information from these areas were collected for a period of seven days of the week to really have a picture of the traffic sizes in the parking sites.

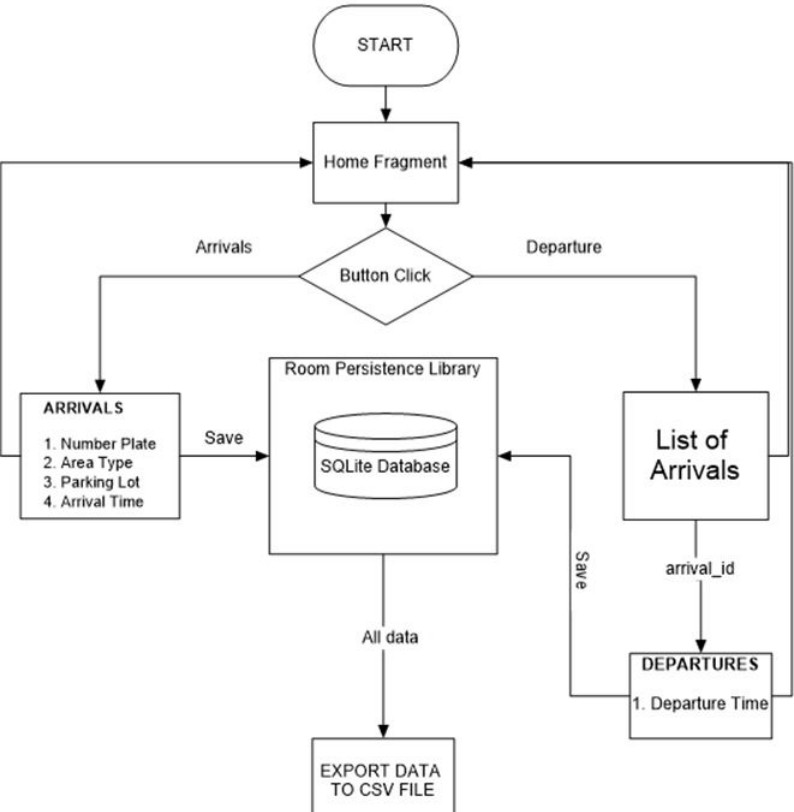

**Figure 6.** Parking management application architecture.

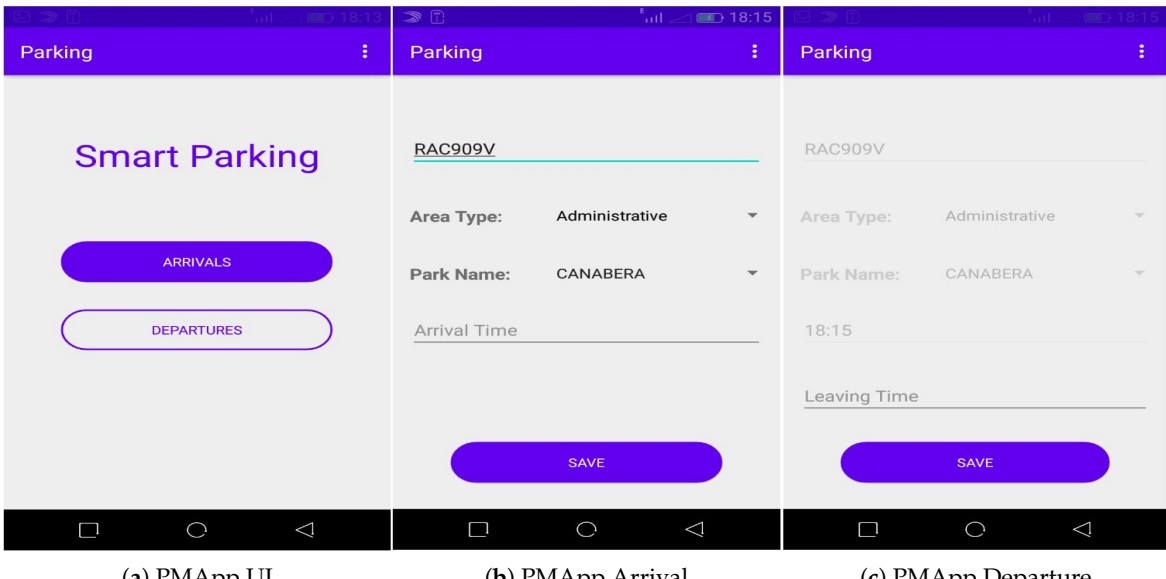

(**a**) PMApp UI  (**b**) PMApp Arrival  (**c**) PMApp Departure

**Figure 7.** Parking Management Application (PMApp).

### 3.4. Weekly Traffic Distribution in the Target Area

The traffic distribution data were gathered based on the three (3) categories; accordingly, we define them as administrative, commercial, and residential, respectively. To have a picture of the daily traffic sizes in each region we observed traffic distribution throughout the week. Figure 8 demonstrates the on-street traffic distribution in the administrative part of Kigali; the area is considered administrative simply because it hosts a big percentage of offices, both public, and private offices are settled in this area. The traffic variation behaves similarly in all regions, morning, and evening hours were observed as the peak hours in terms of traffic distribution. Even if office works are the dominating businesses in this target region, some commercial businesses are carried out as well, and this explains why the traffic distribution shoots up during evening hours.

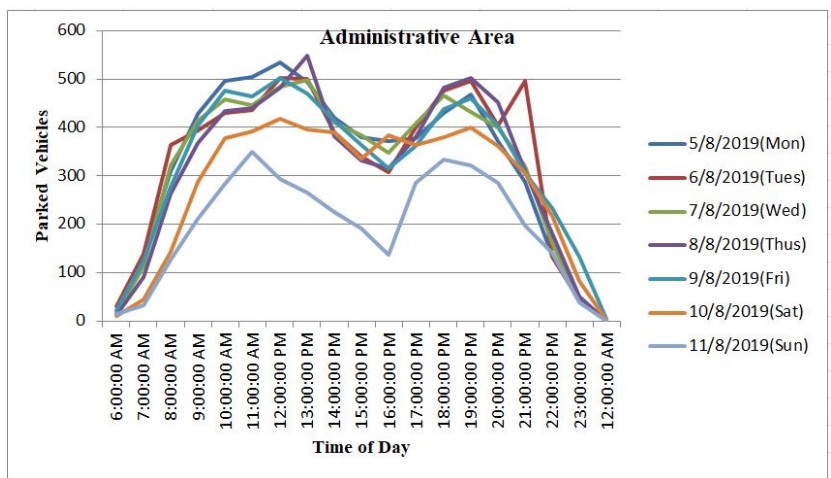

**Figure 8.** Administrative area traffic.

Figure 9 shows the traffic variations in the commercial target area. Since the dominating occupation in the area is commercial, retailing, banking and shopping malls are largely settled in this part of the city, hence we categorize it as a commercial area. The peak hours are observed during the day, especially 10:00–14:00 Hours when commercial business is actively taking place, and later in the evening when people leave offices and visit some shopping malls, the traffic diminishes

completely at midi-nigh, each day of the week. Figure 10 describes the on-street parking traffics in the residential part of the target area where very few parking areas are observed and a few traffic distribution as well. Instead, off-street parking is common in this area. All residents have their vehicles in their home compounds, and very few cars are parked on-street. The scope of this study however is limited to the known on-street parking areas. Parking information was gathered for one week. Whilst few traffic distribution are observed during weekend days, they are still enough to support the roadside gateways. Considering the three urban area types, administrative, commercial, and residential allows us to conduct a spatial-temporal study, the pa distributed computational resources. Furthermore, the parked vehicles could be considered as the unused distributed storage. Exploiting these rich resources will serve as a foundation that is an addition to other global efforts striving to find long-lasting urban vehicular network challenges that all world countries currently face. To optimally exploit such distributed resources, in the next section we introduce vehicle parking shifting models in the aspects of urban areas. We assumed vehicles would be changing sites in the city scenarios, and in each case, the parked vehicle was involved in communication for a specific percentage of its parking period. To make it more meaningful, we considered vehicles that parked for some amount of time, continuing vehicles were not considered.

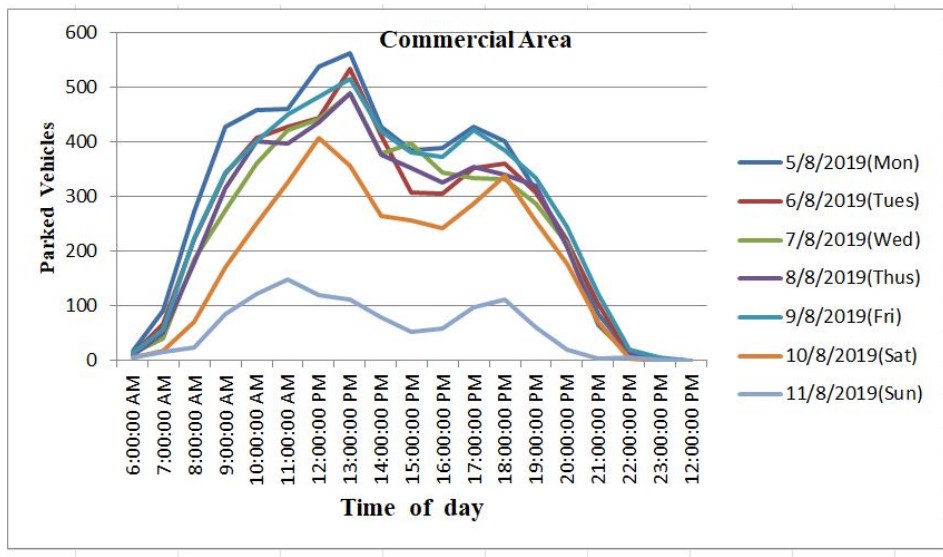

**Figure 9.** Commercial area traffic.

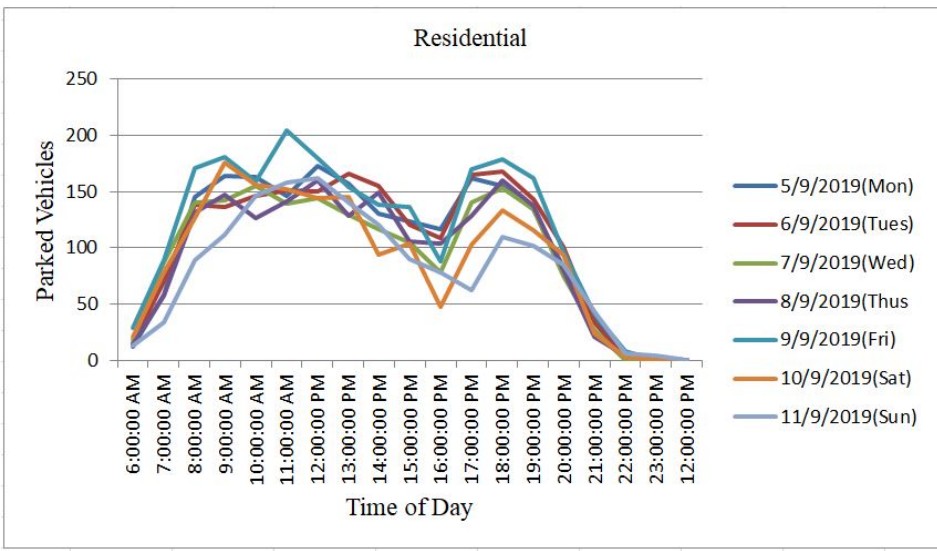

**Figure 10.** Residential area traffic.

## 4. Parked Vehicle Relocation Modeling

In this section, we model a possible city scenario where a single mobile vehicle parks in various lots in the city. The motivation behind this work was escalated after clearly observing the parking dynamism nature in both commercial and administrative areas of interest. In each packing area, we make sure that a specific percentage of parked vehicles take part in gateway duties during a specific percentage of their parking time. In a more intelligent manner, vehicles could among themselves transfer duties to their fellows, when their participation time expires, or when they are leaving a specific parking area.

### 4.1. Vehicle Relocation Modeling and Problem Formulation

For instance, in Figure 11, we demonstrate an area with numerous on-street parking lots where a single vehicle may park in more than one parking area. Our interest is to minimize the communication cost which signifies the number of parked vehicles that actively participate in stationing gateways that receive data from moving vehicles. Interestingly not every time the gateways in the stationary vehicles need to be actively in operation. we consider a situation where a specific percentage of parked vehicles involve in city connection for a certain percentage of their parking periods.

This strategy not only saves the communication bandwidth spectrum but also ensures efficient utilization of the battery and/or any other backup power source supply. In the next subsection, we model the parking shift of the vehicles, and objectively minimize the number of stationary vehicle while communication requirements are sustained. An intelligent application that easily manages the parking time of each vehicle was designed and implemented during this study, to sort out the manual registration of them. The Android studio platform is used to allow seamless development of the parking management application, and this data is structurally stored for modeling purposes.

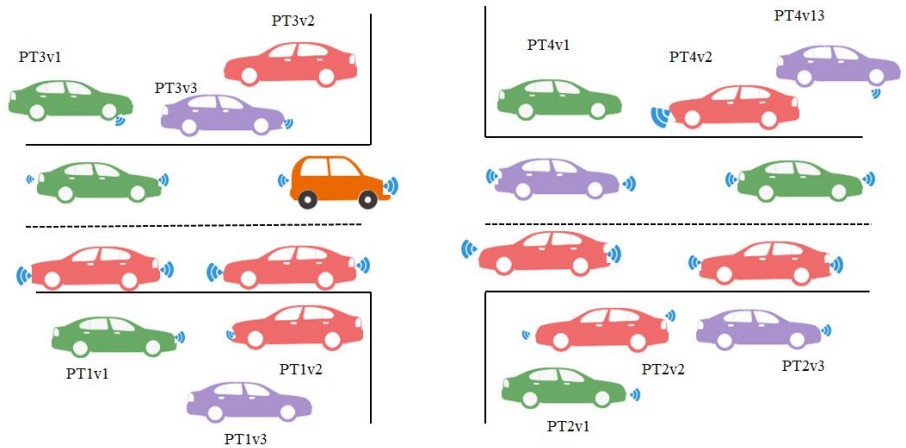

**Figure 11.** Parking Shift Demonstration.

In any parking lot *i*, the parking period of any vehicle m $Pt_{i_m}$ will be given by

$$Pt_{i_m} = Pt_{leav_i} - Pt_{arr}, \tag{1}$$

where:
$Pt_{arr}$ = Is the arrival time of the vehicle in a parking lot
$Pt_{leav_i}$ = Is the time at which the same vehicle leaves the Parking area

### 4.2. Vehicles Parking Matrix

We define a Parking matrix $P_{Matrix}$ to trace the parking duration of every vehicle in each parking site of the smart city scenario. This is an important constraint in our parked vehicle communication model. It determines the possible time that a vehicle will be in a loop of where a temporal gateway would be selected.

$$
P_{Matrix} = \begin{array}{c} \\ V_1 \\ V_2 \\ V_3 \\ . \\ . \\ . \\ V_N \end{array} \begin{array}{cccccc} P_1 & P_2 & P_3 & \cdot \ \cdot & \cdot & P_N \\ \left[ \begin{array}{cccccc} Pt_{P_1V_1} & Pt_{P_2V_1} & Pt_{P_3V_1} & \cdot \ \cdot & Pt_{P_{N-1}V_1} & Pt_{P_NV_1} \\ Pt_{P_1V_2} & Pt_{P_2V_2} & Pt_{P_3V_2} & \cdot \ \cdot & Pt_{P_{N-1}V_2} & Pt_{P_NV_2} \\ Pt_{P_1V_3} & Pt_{P_2V_3} & Pt_{P_3V_3} & \cdot \ \cdot & Pt_{P_{N-1}V_3} & Pt_{P_NV_3} \\ . & . & . & . \ . & . & . \\ . & . & . & . \ . & . & . \\ . & . & . & . \ . & . & . \\ Pt_{P_1V_N} & Pt_{P_2V_N} & Pt_{P_3V_N} & \cdot \ \cdot & Pt_{P_{N-1}V_N} & Pt_{P_{N-1}V_N} \end{array} \right] \end{array}.
$$

The parking matrix technique is employed to determine the total parking time of all parked vehicles under consideration. Most importantly, it helps us to determine how long each vehicle parks in a specific parking lot which helps us to determine the length of time a vehicle participates in supporting the gateway device. In the following equations, we present the total parking time of vehicles $v_1$, $v_2$, $v_3$, ..., $v_n$. The accumulative parking time of each vehicle is realized from all parking lots the vehicle visits, and the individual parking times are summed to get the total parking time of the vehicle. The parking time is a key constraint for our model to determine the percentage of time a certain vehicle spends with an active gateway device. Problem formulation, and gateway stack prototype are presented in the next sections. Optimal solutions are obtained using optimization solvers. The solutions give the number of optimal parking lots and their geographical positions.

$$
\& Pttot_{v1} = \left[ Pt_{P_1V_1} + Pt_{P_2V_1} + Pt_{P_3V_1} + ... + Pt_{P_{N-1}V_1} + Pt_{P_NV_1} \right]. \tag{2}
$$

In Equation (2) above, $Pttot_{v1}$ represents the total parking time of vehicle $v_1$ that is obtained through summing the parking time the vehicle parks in each individual parking sites, for instance $Pt_{P_1V_1}$, $Pt_{P_2V_1}$, $Pt_{P_3V_1}$, $Pt_{P_NV_1}$, are the parking periods of vehicle $v_1$ in the parking areas, $p_1$, $p_2$, $p_3$, $p_N$. Where N stands for the number of parking areas, vehicle $V_1$ visits.

$$
Pttot_{v2} = \left[ Pt_{P_1V_2} + Pt_{P_2V_2} + Pt_{P_3V_2} + ... + Pt_{P_{N-1}V_2} + Pt_{P_NV_2} \right]. \tag{3}
$$

Equation (3) $Pttot_{v2}$ highlights the overall parking duration of vehicle $V_2$; as pointed out above, its total parking duration is given by the summation of the parking times in different parking areas. $Pt_{P_1V_2}$, $Pt_{P_2V_2}$, $Pt_{P_3V_2}$, $Pt_{P_NV_2}$

$$
Pttot_{v3} = \left[ Pt_{P_1V_3} + Pt_{P_2V_3} + Pt_{P_3V_3} + ... + Pt_{P_{N-1}V_3} + Pt_{P_NV_3} \right]. \tag{4}
$$

Equation (4), $Pttot_{v3}$ presents the entire parking time of a vehicle $v_3$ that is computed from the parking duration the vehicle spends in the individual parking lots. Likewise, Equation (5) introduces the total parking time of the $N^{th}$ vehicle. Where all parking time slots the $N^{th}$ vehicle spends in separate parking areas are added together.

$$
Pttot_{vN} = \left[ Pt_{P_1V_N} + Pt_{P_2V_N} + Pt_{P_3V_N} + ... + Pt_{P_{N-1}V_N} + Pt_{P_NV_N} \right]. \tag{5}
$$

### 4.3. Linear Integer Program Formulation

In this section, we consider the parking areas and transmission power of the gateways as constrained resources specifically for small scaled urban areas. The intent of this model is to optimize the utilization of parked vehicles as resources. We minimize the number of parked vehicles in terms of parking areas for gateways stationing, and minimize the gateway power levels. Furthermore, to reduce the deployment costs and terrain utilized for vehicle communication infrastructure deployment, we have modeled the parked vehicles as the intelligent temporal city infrastructure which would be utilized to connect their peers in the reachable proximity. We initiate a novel deployment strategy that ensures $\gamma$ % of the parked vehicles to actively engage in connecting other mobile vehicles for $\zeta$ % of their parking periods. The percentage of the gateway power is as well to be minimized, $\beta$ is the available transmission power power. Further more, to enable our gateways to engage more computational services. In that sense all mobile vehicles passing these parking areas are able to send data through the gateways, each data transmission is considered as a service and requires computational time on the gateway. The static distribution of vehicles is an acceptable assumption, especially in the aspects of urban scenarios. In the following section, we introduce a cost-effective gateway prototype that is built based on lora communication protocol.

$$\text{Min} \sum_{i \epsilon M} \sum_{j \epsilon N} X_i y_{ij} + \sum_{k \epsilon B} \sum_{i \epsilon M} r_k z_{ik} \tag{6}$$

$$\text{s.t} \tag{7}$$

$$\sum_{i=1}^{M} (Pt_{i_m} / Pt_{total_m}) y_i \geq \zeta v_m \ \forall m \ \epsilon \ N \tag{8}$$

$$\sum_{m=1}^{N} v_m \geq \gamma |N| \ \ \forall m \ \epsilon \ N \tag{9}$$

$$\sum_{i=1}^{m} c_i z_{ik} \geq \beta \sum_{i=1}^{m} p_i \ \forall m \ \epsilon \ N \tag{10}$$

$$y_{ij} = \begin{cases} 1 & \text{if a vehicle j is holds a gateway in area i} \\ 0 & \text{otherwise} \end{cases} \tag{11}$$

$$z_{ik} = \begin{cases} 1 & \text{if a vehicle in area i holds a gateway with power level k} \\ 0 & \text{otherwise} \end{cases} \tag{12}$$

$$v_m = \begin{cases} 1 & \text{if a vehicle m respects Equation (8)} \\ 0 & \text{otherwise.} \end{cases} \tag{13}$$

Equation (6) is our objective function that minimizes the required parked vehicles in terms of parking lots, and gateway power level in a city case subjected to the parking vehicles and parking periods constraints. For instance, Equation (8) ensures that if vehicle m is the best candidate to connect others in a self-organizing communication, it does so for the required % of its parking period. Whilst Equation (9) allows a specific percentage of the parked vehicles to engage in such network paradigm, Equation (10) ensures that the desired percentage of the power level is satisfied.

In Table 1, we describe the parameters used in the model. Where as in the next subsection, we describe the LIP problem solutions computed based on the target regions under study. Prior knowledge on the total number of potential parking areas in all three regions is gathered, and their estimated geographical positions are known as well in both open-street map, and simulation of urban mobility. Since the gateway is build based LoRa protocol, it supports long-range communications, with low power consumption. Our gateway prototype can communicate to moving nodes that are situated in a range of more than 10 km. This implies that all moving vehicles in its proximity

and beyond could transmit data through this gateway. Since we are dealing with a network paradigm that involves a massive number of communicating nodes, and voluminous data, there is a possibility that our gateway could run out of computation burst times required to process incoming services. This explains why application servers, data centers, at cloud could be introduced, to handle computation complexity, latency, and bandwidth issues that may arise in vehicular networks. In the next subsection, we present the optimal solutions from the optimization studio for all three types of areas under consideration.

**Table 1.** Model parameters, and their Descriptions.

| Parameter Symbol/Name | Parameter Description |
|---|---|
| $X_i$ | Stands for the $i^{th}$ parking area |
| M | Stands for the number of parking areas available |
| N | Stands for the instantaneous number of available parked vehicles |
| B | A set of gateway transmission levels |
| $\zeta$ | The percentage of time that a parked vehicle is an active gateway (%) |
| $\gamma$ | The percentage of parked vehicles that take gateway duties (%) |
| $Pt_{total\,m}$ | Total Parking time of a vehicle $m$ (seconds) |
| $r_k$ | The cost multiplier of power level based on the number of Gateways |
| $z_{ik}$ | The binary variable of a gateway in area $i$ with power level $k$ |
| $\beta$ | The percentage of the required transmission power (%) |
| $Pt_{im}$ | The parking time a vehicle $m$ spends in parking Lot $i$ (seconds) |
| $v_m$ | A binary variable that is 1 if vehicle m is part of the solution or 0 otherwise |
| $y_{i,j}$ | A binary variable that is 1 if $j$th vehicle in area i hosts an active gateway or 0 otherwise |
| $c_i$ | The demand at area $i$ as gateway transmission powers |
| $p_i$ | a binary variable that is 1, if itinerary close to gateway in area $i$ is covered, 0 otherwise |

### 4.4. Ibm-Ilog Cplex Objective Solutions

In this section, we highlight the objective solutions, which present the optimal number of parking areas that are better candidates for gateway stationing and allow packets from other vehicles passing by the parking area proximity. The IBM ILOG CPLEX Optimization solver [48] is employed to solve the Linear Integer program problem. For each region, that is, administrative, commercial, and residential, we obtain optimal solutions in terms of parking areas geographical positions that will support gateways positioning with minimized execution run times, while ensuring required computational capacity. Specifically, Figure 12 shows the objective solution that describes the optimal placement of gateways in the administrative region. Eleven parking areas are selected as the optimal strategic positions of gateways that will handle most of the computational tasks from the mobile vehicles. In Figure 13, we demonstrate the objective solution with respect to the commercial parking areas, ten identified optimal parking locations are identified that would ensure full coverage of the area. Whereas Figure 14 presents the potential optimal parking areas in residential part.

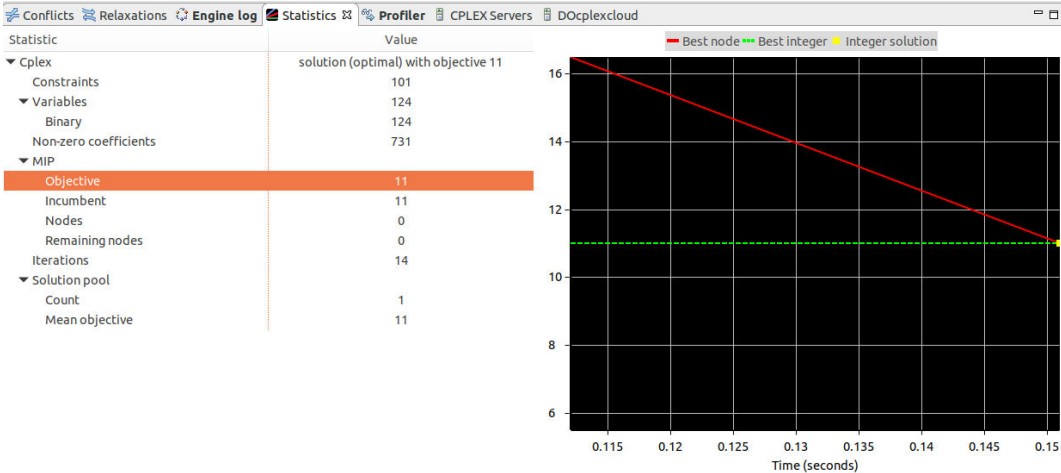

**Figure 12.** Cplex solution: Administrative.

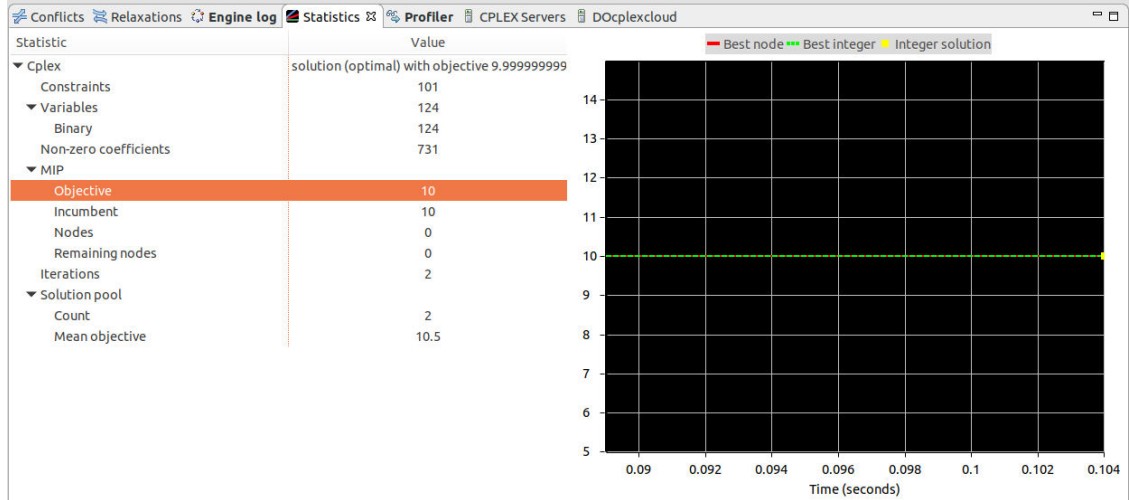

**Figure 13.** Cplex solution: Commercial.

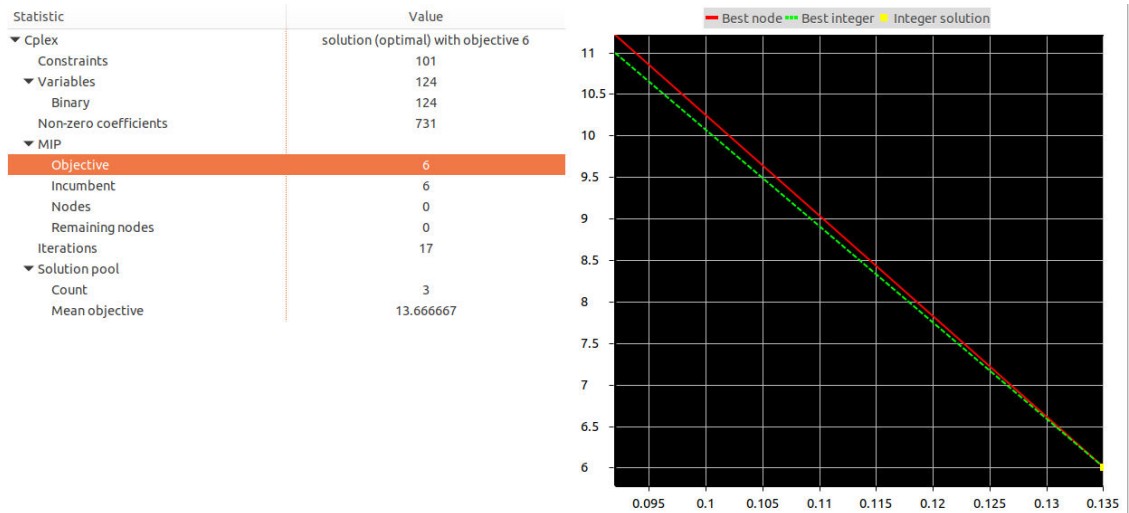

**Figure 14.** Cplex solution: Residential.

For instance, Figures 15 and 16 demonstrate the physical positions of the optimal parking areas returned from the optmization solver. We employ satellite imagery to demonstrate the images of the potential on-street areas that could optimally be good candidates for gateways stationing. Interestingly, the optimal locations are realized from the known city parking sites. While the optimization solver tool is populated with the parking areas, and the average number of parked vehicles in each parking area, we identify the ones that could optimally hold gateways and receive data packets from the mobile vehicles in its proximity. Accordingly, each parking lot is represented by its geographical coordinates, that is, latitude and longitude from the parking areas on the map are utilized to represent the vehicles' positions.

In the next section, we demonstrate the gateway prototype that could support long-range data transmissions from vehicles to vehicles, based on the defined source-destination entities. Experimental results are presented to indicate the power of the prototype to support numerous tasks. As pointed out down, two compliance Lora sensors were employed to test the prototype. The prototype employs MQTT (Message Queuing Telemetry Transport) that utilizes Transport Layer Security, and Secure Socket Layer that initiates a secure communication between a client and a server. Being cryptographic protocols with a handshake procedure, TLS, and SSL ensure secure communication between devices. This guarantees the privacy, and some level of security of the data transmission specifically at the network layer.

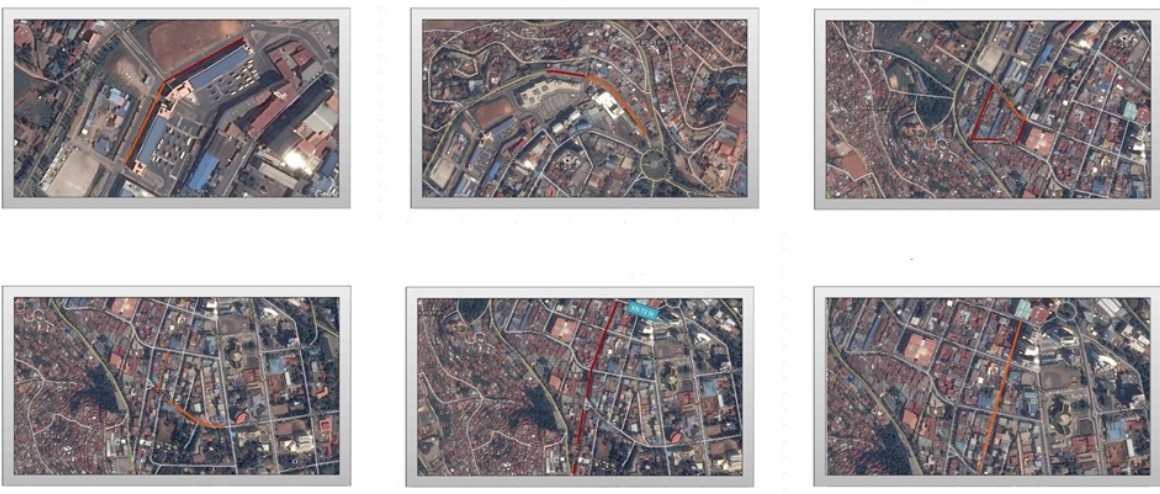

**Figure 15.** Physical Positions of the optimal Parkings (Administrative).

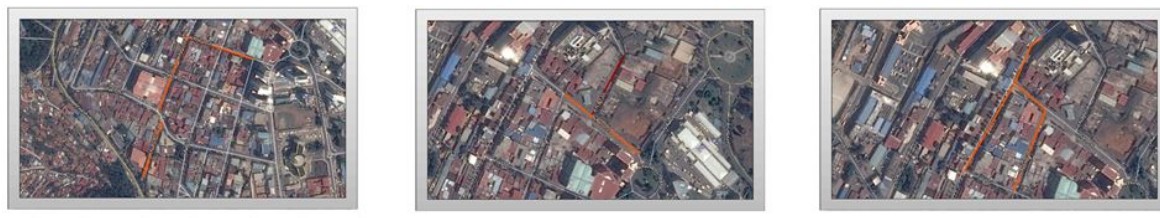

**Figure 16.** Physical Positions of the optimal Parking (Commercial).

## 5. Open Innovation Engineering in Optimal Exploitation of On-Street Parked Vehicles

Open innovation has been in existence for a number of years, and has obtained progressive consideration in scientific research. It employs purposive inflows and outflows of knowledge to speedup internal innovation, and to extend the markets for the external use of innovation. Open innovation is transforming all fields such as engineering [49], sciences, markets, and so forth. Numerous entities such as research institutions, manufacturing Systems, and innovative government projects are engaging in open innovation as well, and diverse insights are observed [50]. While open innovation has been dominating in large high-tech multinational enterprises, it is currently being adopted in small and medium enterprises as well [51]. The business model and open innovation of parked vehicles in this aspect is based on two dimensions, that is, Parking management application (PMApp), and the gateway device deployment. With PMApp both drivers and toll collectors can figure out the length of time a vehicle has been in a specific parking site. Interestingly the application is cloud supported and allows multiple users to access it at a time, which implies that more tolls could be supported at an instant. Not like KakaoT that supports mobile tax drivers to locate the customer's position, navigate to the nearest parking space [52]. PMApp is limited to parking duration management. One hundred Rwandan francs is charged for every hour of parking. The gateway device deployment in on-street parked vehicles is expected to relay important messages to the intended recipients. For instance, information on when the next public bus will arrive at the bus stop is very important to people already at the stage, or for others who need to plan their trips in a smart and precise way. The potential stakeholders that are expected to benefit from the system once deployed are system developers, public transport agencies, passengers, internet providers, and the government agencies in charge of transportation, smart city practitioners.

*Innovative Gateway Prototyping for Vehicle Communication*

In this section, we prototype an envisioned in-vehicle gateway that facilitates socially-related data transmissions from vehicular on-board units to other vehicles, and/or our local application servers. A laboratory-based prototype of a simplest gateway device is presented in this section. Hardware modules incorporated for the prototype are explained here below. The gateway prototype exploits Raspberry Pi as the control unit. A simplified single-board computing device 64-bit, 1.2 GHz ARM Cortex-A53 processor. Its pin headers are compatible with LoRa Raspberry Pi expansion board. The expansion board gives the device both receiving and transmission capabilities through its radio modules, packets received from LoRa enabled sensing nodes could be transmitted to the cloud, data center, and particularly our application server.

The initial prototype suggested in this research could be powered by the vehicles while in motion, and is fitted with a Raspberry pi battery pack with a battery club that powers the gateway while vehicles are parked. Figure 17 describes the block diagram of the lab-based gateway prototype. The power expansion board is constructed in such a way that it is exactly compatible and provides the raspberry pi with power from an attached lithium battery rated with 3.7 V 3800 mAh.

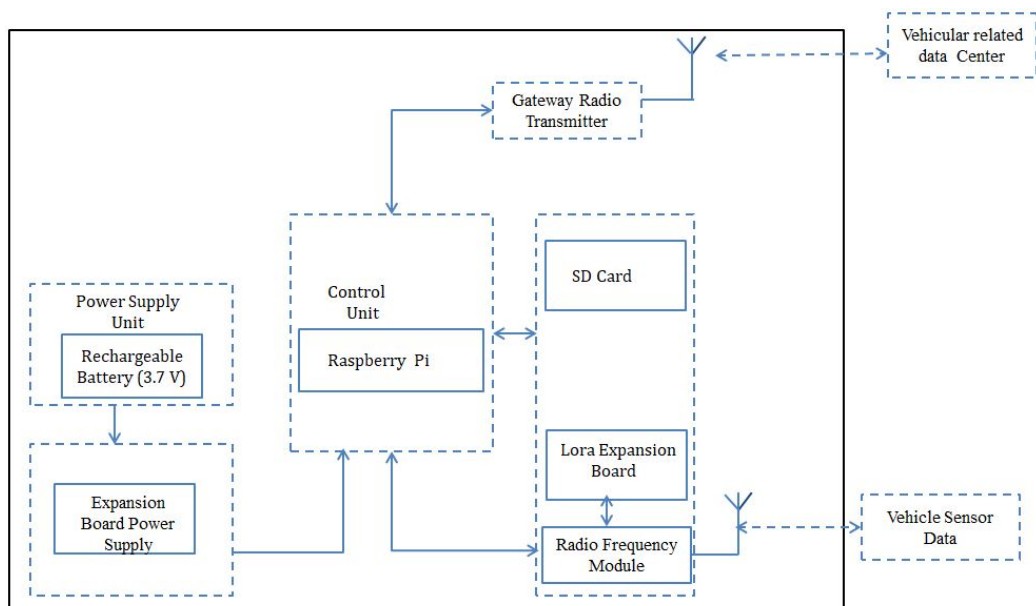

**Figure 17.** Block Diagram of the gateway prototype.

The prototype stack is mainly constructed from three (3) layers, that is, the control unit (CU), the LoRa protocol expansion board, and power supply unit Figure 18a. While there are crucial hardware modules that helped us to construct the prototype, there is one important firmware that is the core heart of the prototype that is, noobs, and the sd card is used to load the firmware into the raspberry pi. Noobs involves a raspberry pi operating system and allows us to fully interact with the gateway. Noobs is simply an operating system installer that has a raspberry pi OS. Figure 18b shows the back end operation of our gateway. Important operations could be performed in a back-end mode.

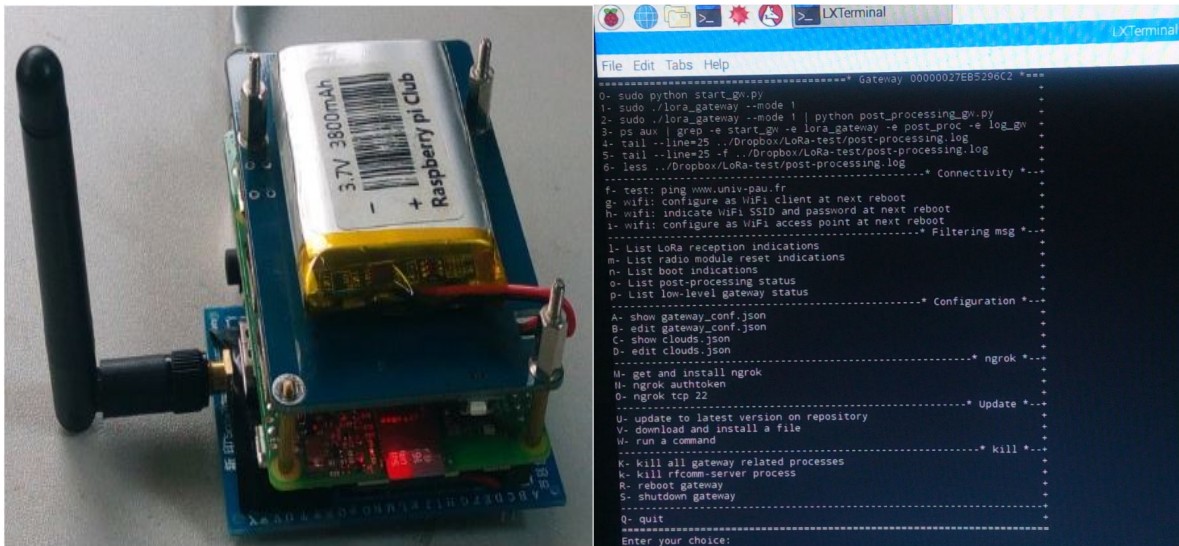

(**a**) Prototype Stack    (**b**) Back-end operations

**Figure 18.** Side view of the prototype stack and back-end operations.

Two Lora enabled sensors are employed to test the prototype and communicate based on the internet of thins three-tier layered architecture. To achieve the testing goal, the LS-111P and LS-113P LoRaWAN compliant sensors are employed to test the prototype the sensors communicate to the gateway through lora protocol. The LoRa based gateway testing is based on three basic layers of the internet of things, the perception layer, where target sensing parameters are captured from the target environments. The operation of our prototype is at the next layer, the networking layer. Packets containing the sensed data are all convergence to the gateway that transmits the data to the cloud that host server applications, data storage, processing could be performed here. Via end-user applications data retrieval and visualization is carried out at this level. The Figure 19, describes the architecture of gateway device testing using LoRa enabled sensors.

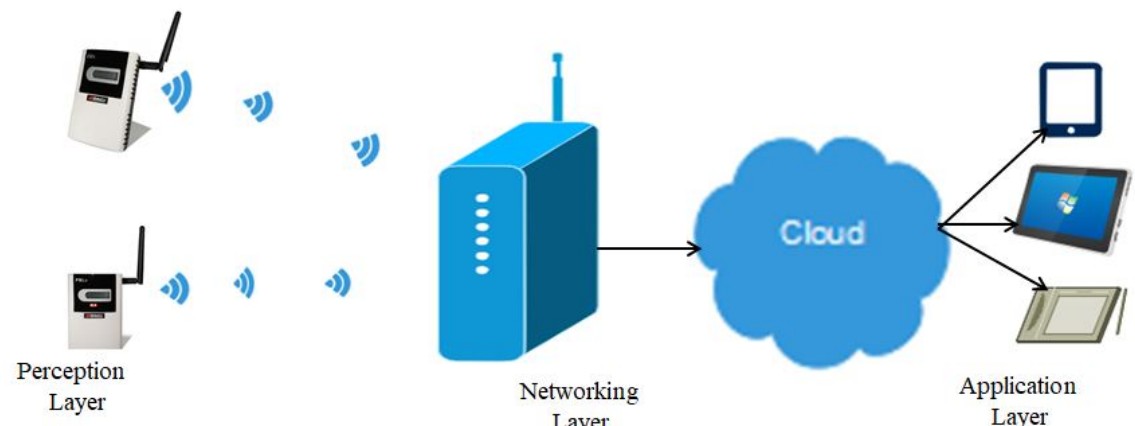

**Figure 19.** System testing using LoRa sensors.

## 6. Experimental and Simulation Results Analysis

Initial lab testing was conducted to evaluate the performance of the prototype and its related services. For each computational task, we also evaluate the processor resource utilization. The operational tests included the sensing parameters and transmitting them to the gateway and uploading them to the application server. The communication between the LoRa sensors and the gateway prototype was conducted using radio frequency communication through the configured

LoRa expansion board module. The initial laboratory testing demonstrated a well-functioning communication system, with application server receiving relevant sensor information from the prototype.

In the diagram Figure 20, we demonstrate the the retrieval of sensor1 information on the application server from the gateway. The sensor unit captures the level of $CO_2$ from in vehicle environment and transmits the sensed information through LoRa radio protocol to the gateway. The sensor unit is compensated with signal status and weather condition sensing capabilities. The retrieved information comprised of multiple dimensions and these include: "TimeStamp", "DeviceName", "ChanleNumber", "SensorName", "SensorID", "SensedParameter", "Unit", "Value".

| Timestamp | Device N... | Channel | Sensor Name | Sensor ID | Data Type | Unit | Values |
|---|---|---|---|---|---|---|---|
| 2020-03-02 5:36:59 | GblSat CO2(3... | 101 | SNR | 11895960-cba0-11e8-bc72-5... | snr | db | 13 |
| 2020-03-02 5:36:59 | GblSat CO2(3... | 100 | RSSI | 14b7dd00-cba0-11e8-a324-1... | rssi | dbm | -101 |
| 2020-03-02 5:36:59 | GblSat CO2(3... | 1 | Humidity | 13961950-cba0-11e8-a7e6-5... | rel_hum | p | 64.96 |
| 2020-03-02 5:36:59 | GblSat CO2(3... | 0 | Temperature | 15bf13d0-cba0-11e8-90bb-9f... | temp | c | 24.44 |
| 2020-03-02 5:36:59 | GblSat CO2(3... | 2 | CO2 | 17894c80-cba0-11e8-bc72-5... | co2 | ppm | 468 |

**Figure 20.** Sensors Information on application server from the gateway prototype: Sensor1.

While in the diagram above Figure 21 we demonstrate the sensed parameters captured from sensor2.

| Timestamp | Device N... | Channel | Sensor Name | Sensor ID | Data Type | Unit | Values |
|---|---|---|---|---|---|---|---|
| 2020-03-02 5:36:41 | GblSat PM2.5... | 100 | RSSI | 7baa6af0-93b1-11e7-9727-5... | rssi | dbm | -94 |
| 2020-03-02 5:36:41 | GblSat PM2.5... | 1 | Humidity | 7c7d9970-93b1-11e7-9727-5... | rel_hum | p | 68.72 |
| 2020-03-02 5:36:41 | GblSat PM2.5... | 101 | SNR | 7c622230-93b1-11e7-9d8b-ff... | snr | db | 15 |
| 2020-03-02 5:36:41 | GblSat PM2.5... | 0 | Temperature | 7c898050-93b1-11e7-a491-d... | temp | c | 23.07 |
| 2020-03-02 5:36:41 | GblSat PM2.5... | 2 | Particle Matter | 7cf07f80-93b1-11e7-9d8b-ffc... | analog_sensor | null | 26 |

End of list

**Figure 21.** Sensors Information on application server from the gateway prototype: Sensor2.

Similar attributes are defined in sensor2 as in sensor1. All sensing devices have compensated capabilities to sense both signal status and weather condition in terms of signal-to-noise (SNR) ration, received signal strength indicator (RSSI), temperature and humidity of the environment. Apart from evaluating the operational status of the prototype in terms of communication links, to realize its capacity for the accommodation of more tasks, we carried out further testing to find out the processor resource utilization of each of the computational tasks Figure 22. The average percentage run time on the processor regarding each computation task is demonstrated. STAT represents the service that keeps track of the run time of all other computational tasks, and it is a task on its own. IDLE presents the status of the gateway prototype system when it has no job to deal with. Sensor1, Sensor2 demonstrates the computational run time of prototype communication links of the two sensors respectively. While APS is the run time of the system when it is uploading the sensor information to the application servers.

The processor resource utilization used by the system components proves that additional sensors, and other computational tasks could be added. More social-related data could be captured and communicated through the gateway. In the next subsection we conduct a simulation study under all three area types under consideration. The simulation studies in our technique assumes deployment of gateways in a specific percentage of the parked vehicles and these will act as temporal on-road infrastructures that support other vehicles. In this simulation study we demonstrate on-street parked

vehicles as potential distributed resources that will benefit urban transportation planners to avail computational, as well as communication infrastructures that will ensure a sustainable city.

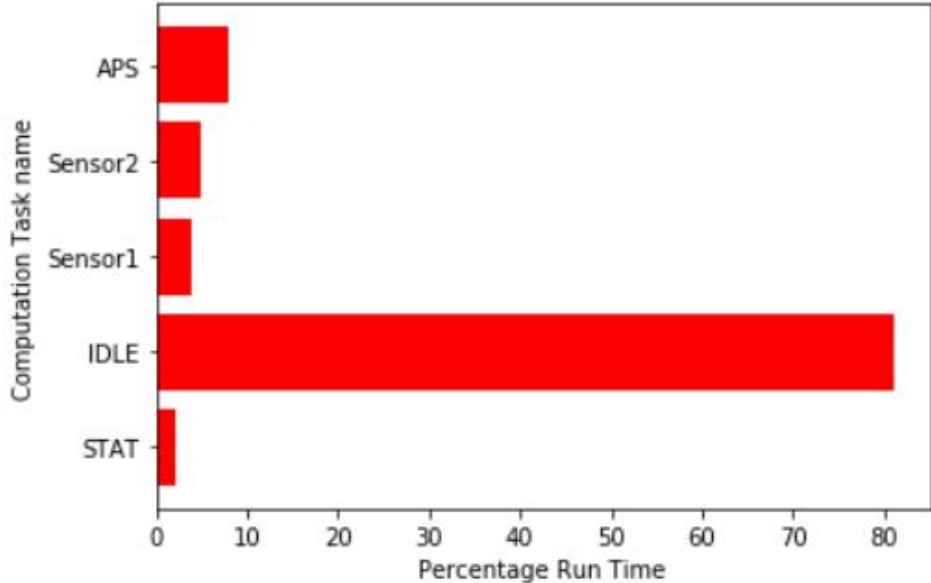

**Figure 22.** Computation task processor utilization.

*6.1. Simulation Setup*

Initially, we carried out a weekly survey on an urban area of Kigali, the capital city of Rwanda. Common on-street parking areas are selected to have a view on the daily average parked vehicle sizes. Weekly real parking information from the indicated sites were collected. For simulation studies, an area of $2000 * 2000$ m is selected. In the next subsection, we carry out simulation studies that assume the deployment of the gateway device in a specific percentage of the parked vehicles. To emphasize the time factor, we enable the candidate vehicles engage in gateway positioning for a specific percentage of their parking duration.

Apart from our proposed approach, two more methods were studied to evaluate the performance of our method. Firstly, an RSUs deployment approach is considered. Secondly, an inter-vehicle communication strategy is simulated and finally, our approach, that is, the on the street parked vehicle approach is learned for comparison purposes. To evaluate the performance of the network in various scenarios, renowned Network Quality of service metrics are evaluated, and for numerous vehicle densities. In the following Table 2, we explain the parameters used in network performance evaluation simulation.

**Table 2.** Simulation Parameters for Network Performance Evaluation.

| Parameter Symbol/Name | Parameter Description |
|---|---|
| Average Parked Vehicles (Administrative, Commercial, Reidential) | 300, 220, 98 |
| Mobile vehicles | 60, 120, 180, 240, 300 |
| Packets Routing Protocol | Ad Hoc Demand Vector (AODV) |
| MAC Layer Protocol | IEEE 802.11p |
| Traffic Generator | Simulation of urban mobility (SUMO) |
| Mobility Model | Long Distance path Loss |
| Antenna height | 3 m |
| Transmission Power | 25 dBm |
| Transmission range | 250 m |
| Network Simulation Tool | Network Simulator version 3 |
| Simulation Time | 100 s |

## 6.2. Network Simulation

In this section, we present a brief discussion of our numerical results. We initially conducted a study that learned the performance of on-street parked vehicles as vehicular roadside temporal infrastructures in all the urban regions under consideration. Figure 23a,b show the success packets delivery, and overall data throughput under different mobile traffic volumes. For comparison purposes, roadside deployment is study as well with all hot-spots road intersection points acting as the RSUs deployment sites, free vehicle-to-vehicle communication study is conducted at the same time. In all simulation instances, parked vehicles yield a good communication performance with higher data throughput, and success packets delivery.

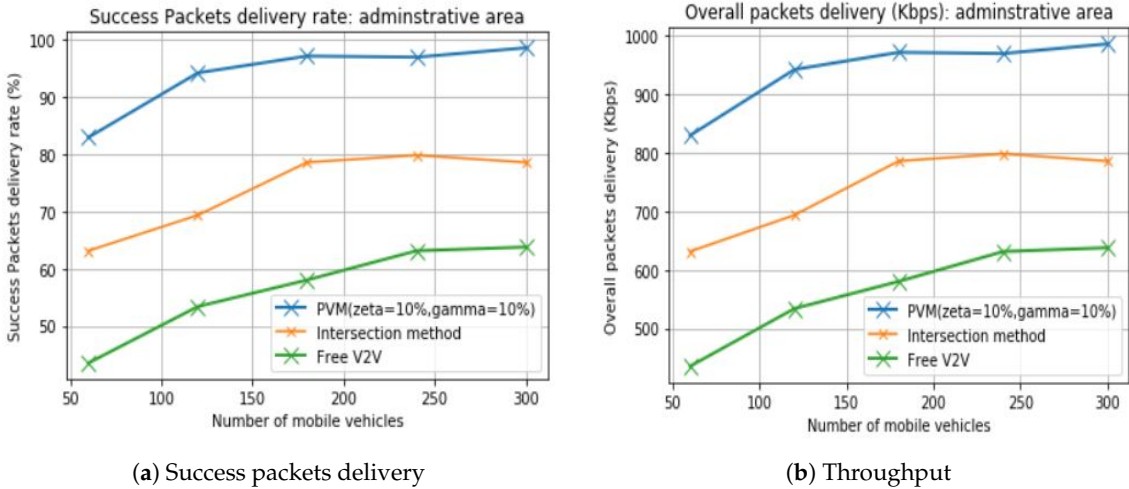

(**a**) Success packets delivery (**b**) Throughput

**Figure 23.** Overall packets delivery & Throughput: Administrative area.

While Figure 24a,b demonstrate the packets delivery ratio, and overall throughput in the commercial area of consideration. It is well noted that in all approaches, parked vehicle communication outperforms others. Interestingly, a small percentage of the parked vehicles is considered. Every single instance utilizes 10% of the available parked vehicles.

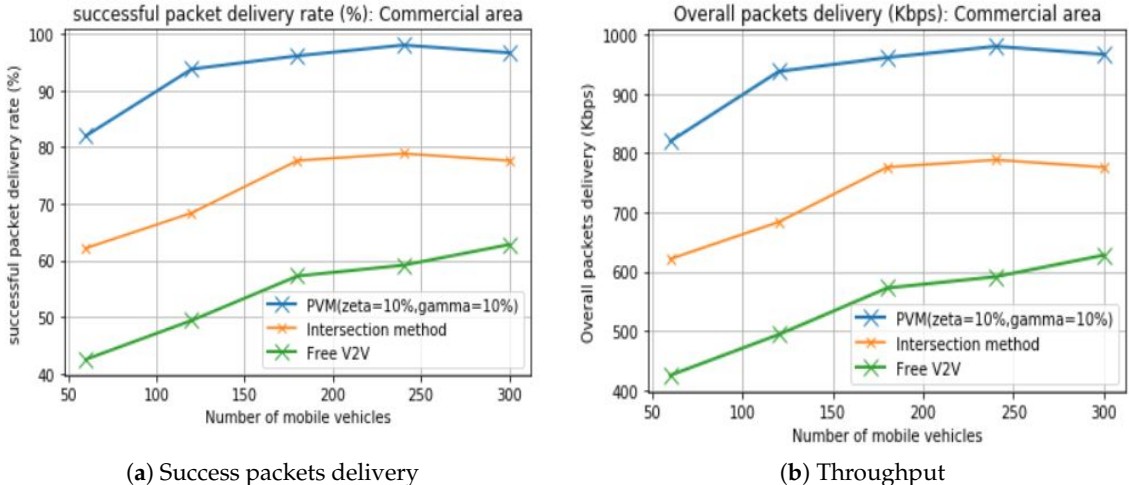

(**a**) Success packets delivery (**b**) Throughput

**Figure 24.** Success Packets delivery & Throughput: Commercial area.

Furthermore, Figure 25a,b indicates the performance effectiveness evaluation in residential areas which also presents parked vehicles as the best temporal roadside infrastructures, that could supplement mobile vehicular communications. In all instances, vehicle-to-vehicle communication presents low packets delivery success, simply because excessive communication links break between vehicles that are not in each others' communication range. Packets delivery success rate counts for

the loss of packets that do not successfully reach the intended destination. The same issue could be resolved by the several parked vehicles to handle over the packets to the target recipients. Several other technical communication challenges are resolved, such as bandwidth consumption in re-transmitting unsuccessful packets, battery overuse due to the re-sending of packets. Benefiting distributed resources in the parked vehicles will turn the smart city development from dreams to reality, especially in the aspects of computation, storage, and low latency communication. Edge devices could be located on the parked vehicles and support on-road mobile vehicles. Based on the best performance of the low percentage employed in our study, once all vehicles are equipped with the gateways, they will be able to support a greater number of vehicles than the actual available mobile vehicles. Urban transport may adopt this deployment strategy to decide which vehicles could serve as the best candidate for Social information delivery.

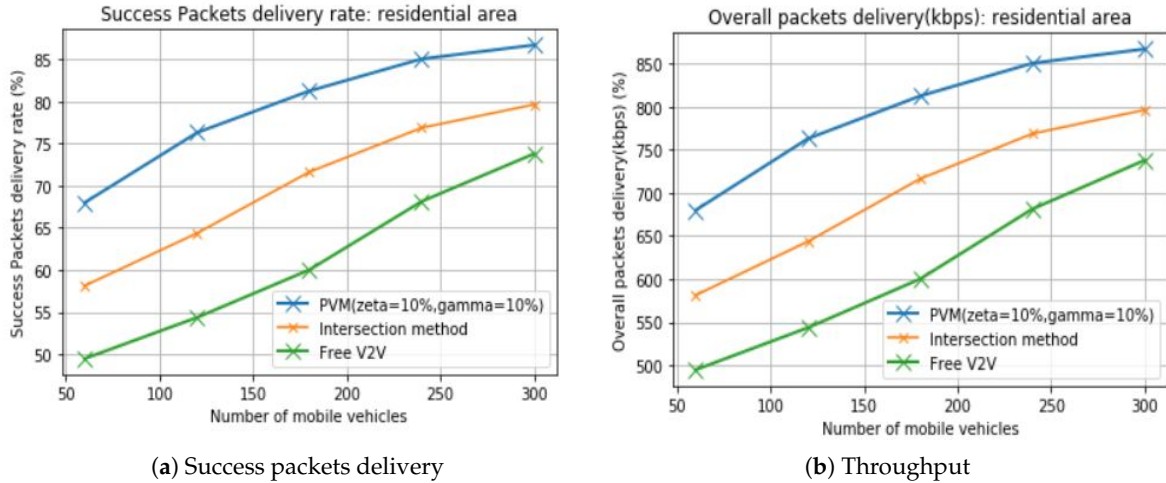

(**a**) Success packets delivery  (**b**) Throughput

**Figure 25.** Success Packets delivery & Throughput: Residential area.

From the observation, for all three considered areas, the numbers of parked vehicles are greater than that of mobile vehicles, especially in both administrative and commercial regions. This explains why utilizing parked vehicles as resources is very crucial. For comparison purposes, we learn other communication strategies that are discussed here below.

1. Intersection-Based Deployment. RSUs are placed at the identified potential road intersections, that are selected for deployment reasons in vehicular communications. To make more sense of comparison, we consider the same transmission powers of RSUs, and units used by the parked vehicles for simulation purposes. In all areas under consideration, potential road intersections are identified, and RSUs deployed based on their geographical positions.

2. Parked Vehicle Approach. In this modeling technique, access points are assumed in a specific percentage of vehicles for a specific percentage of their parking periods. The traffic mobile volumes are generated using SUMO. In simulations, carried out in this study, for each optimal parking lot we consider only 10% parking vehicles for 10% of their parking periods. To ensure moderate utilization of the parked vehicles. We consider vehicles that parked at least a minimum of 100 s, and not beyond 30 min.

3. Free Vehicle to Vehicle. In this method, mobile vehicles communicate among themselves, without being aided by any infrastructure. In the previous communication approaches, uplink, downlink communications are considered between vehicles, and infrastructures, and infrastructures to vehicles, respectively.

*6.3. The Impact of Traffic Volumes*

By analyzing the three communication approaches, it was observed that the communication becomes better in terms of throughput, and packets success delivery rate when the traffic volumes

of the mobile vehicles also rise. This is true for all kinds of areas under consideration because vehicles theme-selves relay messages to their peers. Moving vehicles transmit the packets to the intended recipients in their proximity. Even though, roadside infrastructures take up this role. But the cooperation of roadside infrastructures (Both parked vehicle, and Roadside units) and mobile vehicles presents better results. To ensure homogeneity of the system components, we assumed the deployment of RSUs made from the same hardware have the same operational capacity in terms of computation, storage and similar transmission power, and transmission ranges. Specifically, our prototype ensures good coverage, since it has a good transmission range of several kilometers (over 10 km). The prototype assumed the parked vehicles could also be generically utilized for other various applications in the aspects of smart city development. Simulation observations have indicated that packets' delivery successes are greatly affected by the vehicles' speeds; with higher speeds, several communication links break. Moreover, collecting massive data from many mobile vehicles has become a big challenge, especially in the aspects of data privacy. Intelligent algorithms, and protocols that do allow vehicles to seamlessly communicate without communication links' deterioration based on the vehicles' mobility are equally crucial, most importantly data privacy should be ensured, this shapes our future direction of research.

## 7. Conclusions and Future Research

In this study, we optimally exploited the parked vehicles as roadside gateways for the social internet of vehicles. We demonstrate parking vehicles as computational resources that are underutilized in the areas of study. An integer linear program (ILP) problem was formulated and solved using an optimization solver. The results enable a specific percentage of on-street parked vehicles to act as roadside gateways for a specific percentage of their parking periods. Network performance evaluation was carried out through simulation studies. Since the parking period of each vehicle is a factor of concern, we also designed and developed an android based mobile application, a Parking management application (PMapp) that smartly determines the parking time of each vehicle under consideration. The numerical results show that on-street parked vehicles yield enough computational resources that could take up the duties of road side gateways temporarily. Furthermore, a cost-effective gateway prototype based on the LoRa technology is produced that presents extended computational run time that could accommodate more computational tasks, and requests from distant mobile vehicles could be attended. While the scope of this research is limited to the optimal exploitation of on-street parked vehicles as gateways in the study area, and making sure a specific percentage of the available vehicles engage in social communication for a specific percentage of their parking duration, handling data from multiple vehicular IoT devices, and ensuring their privacy becomes a great concern. This therefore shapes our future research direction. Handling voluminous data from urban vehicular IoT devices, their mobility context-aware representation and ensuring the data privacy would be our future research focus.

**Author Contributions:** Idealization, Data Collection, Modeling, simulation, Device prototype and testing, Initial Draft preparation, T.E.; Application Development, and its documentation,T.E., W.K.; Draft proof-reading, co-supervision, J.R., Overall Work Supervision, R.D. All authors have read and agreed to the final draft of this manuscript.

**Funding:** This study was under a direct financial support of the African Center of Excellence in the Internet of Things(ACEIoT)-Under the college of Science and Technology-University of Rwanda.

**Acknowledgments:** The authors would like to extend their acknowledgments to the African center of excellence in the Internet of Things-University of Rwanda for the provision of required components for the prototype. The contribution of former Kigali Veterans Cooperative Society now Millennium Savings and Investment Cooperative, especially their tolls collection agents for entire week traffic data collection from the target area.

**Conflicts of Interest:** The authors declare no conflict of interest.

**Abbreviations**

The following abbreviations are used in this manuscript:

| | |
|---|---|
| SIoV | Social Internet of Vehicles |
| IoV | Internet of Vehicles |
| IoT | Internet of Things |
| VANET | Vehicular Add Hoc Network |
| RSU | Roadside unit |
| V2X | Vehicle to Everything |
| V2D | Vehicle to Device |
| V2G | Vehicle to Grid |
| PMApp | Parking management Calculation App |
| VANET | Vehicular Add Hoc Network |
| ILP | Integer Linear Program |
| SUMO | Simulation of Urban Mobility |
| OSM | Open-Street Map |
| WAVE | Wireless Access Vehicular Environment |
| ILP | Integer Linear Program |

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
