# Peer review of "Optimal Exploitation of On-Street Parked Vehicles as Roadside Gateways for Social IoV—A Case of Kigali City"

_2199-8531, doi:10.3390/joitmc6030073_

Round 1

Reviewer 1 Report

Dear author(s),

I send you my review in the attached file.

Reviewer 2 Report

The paper deals with an interesting topic and provides a good basis for interesting future research. The paper is original, written in good English, with appropriate methodology and theoretical background as well as good literature review. The structure of paper is logical, text is easy to read. The findings are presented and discussed. The aim of the paper was to exploit the parked vehicles as roadside gateways for social internet of vehicles. Authors demonstrated parking vehicles as computational resources that are under utilized in the areas of study. An integer linear program (ILP) problem was formulated and solved using an optimization solver.  It should be taken into account that the research still has some limitations, however, this section is not described by authors. Please add the section about the limitations as well as potential future areas of the research.

Reviewer 3 Report

The internet of things and the social aspects of vehicles are considered in this paper towards building internet of vehicular things (IoVT). A mobile application has implemented to manage the parking duration of the vehicles. Two performance metrics have been compared - packets success delivery rate and overall packets throughput. The social IoVT aspect is also has been considered and the authors argued that - "security of nodes participating in social internet of things has been a great concern". The paper also talks about cyber entities and security services. Another concern in this paper is 'accessing vehicular environments' and 'the role of roadside access points to allow vehicles with common social interest exchange messages'.

The contributions seem well presented along with reference scenario, architecture, methodology, simulation design, parking prototype application and evaluation. The related works are well covered.

However, the authors can have a look at the direction of 'accessing data from multiple IoT sources', like multiple vehicular IoT sources, how to manage them and how to handle privacy when data are coming from multiple SIoV sources. The following papers can have a look, although these are not in the vehicular direction, in the general IoT direction. The authors can add a 'future work' focusing both multiple IoVT sources and data privacy. The last section can be named as 'Conclusion and Future Research'.

Ref: IoT Streaming Data Integration from Multiple Sources, Computing journal, 2020; Integration of IoT Streaming Data with Efficient Indexing and Storage Optimization, IEEE Access, 2020; Accessing Data from Multiple Sources Through Context-Aware Access Control, IEEE TrustCom, 2018.
